# Casein Kinase 2 Alpha Inhibition Protects against Sepsis-Induced Acute Kidney Injury

**DOI:** 10.3390/ijms24129783

**Published:** 2023-06-06

**Authors:** Jeung-Hyun Koo, Hwang Chan Yu, Seonhwa Nam, Dong-Chan Kim, Jun Ho Lee

**Affiliations:** 1Department of Biochemistry and Molecular Biology, Jeonbuk National University Medical School, Jeonju 54896, Republic of Korea; jhyuni81@jbnu.ac.kr (J.-H.K.); ghkdcksfks@gmail.com (H.C.Y.); 2Department of Anesthesiology and Pain Medicine, Jeonbuk National University Medical School and Hospital, Jeonju 54896, Republic of Korea; dd3403@naver.com (S.N.); dckim@jbnu.ac.kr (D.-C.K.); 3Research Institute of Clinical Medicine of Jeonbuk National University-Biomedical Research Institute of Jeonbuk National University Hospital, Jeonju 54896, Republic of Korea

**Keywords:** sepsis, acute kidney injury, CK2α, TBBt, NF-κB

## Abstract

Sepsis-induced acute kidney injury (AKI) is a common complication in critically ill patients, often resulting in high rates of morbidity and mortality. Previous studies have demonstrated the effectiveness of casein kinase 2 alpha (CK2α) inhibition in ameliorating ischemia–reperfusion-induced AKI. In this study, our aim was to investigate the potential of the selective CK2α inhibitor, 4,5,6,7-tetrabromobenzotriazole (TBBt), in the context of sepsis-induced AKI. To assess this, we initially confirmed an upregulation of CK2α expression following a cecum ligation and puncture (CLP) procedure in mice. Subsequently, TBBt was administered to a group of mice prior to CLP, and their outcomes were compared to those of sham mice. The results revealed that, following CLP, the mice exhibited typical sepsis-associated patterns of AKI, characterized by reduced renal function (evidenced by elevated blood urea nitrogen and creatinine levels), renal damage, and inflammation (indicated by increased tubular injury score, pro-inflammatory cytokine levels, and apoptosis index). However, mice treated with TBBt demonstrated fewer of these changes, and their renal function and architecture remained comparable to that of the sham mice. The anti-inflammatory and anti-apoptotic properties of TBBt are believed to be associated with the inactivation of the mitogen-activated protein kinase (MAPK) and nuclear factor κB (NF-κB) signaling pathways. In conclusion, these findings suggest that inhibiting CK2α could be a promising therapeutic strategy for treating sepsis-induced AKI.

## 1. Introduction

Sepsis, a severe systemic inflammatory disease with high mortality, is caused by dysregulated host response to bacterial infection [1]. Multiple organ failure is a major complication that leads to high mortality in patients with sepsis [2]. When acute kidney injury (AKI) occurs during or after sepsis, it increases the mortality rate by approximately 70% [3]. Of note, levels of circulating inflammatory cytokines such as tumor necrosis factor-α (TNF-α) and interleukin (IL)-6 are positively associated with the increased risk of mortality in AKI patients [4,5]. It is widely accepted that lipopolysaccharide (LPS) released from Gram-negative bacteria binds to Toll-like receptor 4 (TLR4) and transduces signals to activate several signaling pathways, including mitogen-activated protein kinases (MAPKs) and nuclear factor κB (NF-κB). These pathways ultimately increase transcription of pro-inflammatory cytokines and other inflammatory mediators [6]. Despite the growing understanding of the pathophysiological processes of sepsis-induced AKI, however, the treatment of septic AKI has still been unsatisfactory.

Casein kinase 2 (CK2), also known as protein kinase 2, is a protein serine/threonine kinase; it is a tetrameric enzyme composed of two catalytic subunits (α and/or α′) and two regulatory beta subunits [7]. A large number of studies show that CK2 has been found to phosphorylate various transcription factors regulating the inflammatory response, including NF-κB [8]. Specifically, CK2 phosphorylates inhibitory κB (IκB)α and degrades it through the proteasomal pathway [9]. In addition to IκBα, CK2 also directly phosphorylates p65, thereby amplifying its transcriptional activity [10]. Overall, inhibition of CK2 activity implicates attenuating inflammation and cytokine signaling via the suppression of NF-κB.

Over the past two decades, numerous inhibitors of CK2, such as emodin, apigenin, and 4,5,6,7-tetrabromobenzotriazole (TBBt) have been discovered and developed. Among these inhibitors, TBBt is the most effective cell-permeant inhibitor of CK2. TBBt selectivity is obtained via a hydrophobic pocket adjacent to the ATP/GTP binding site, which is smaller in CK2 than in other protein kinases [11]. Ka et al. previously showed that TBBt is useful to ameliorate ischemia–reperfusion-induced AKI [12]. Mechanistically, the renoprotective effects of TBBt were associated with the suppression of MAPKs and NF-κB pathways. Because MAPK and NF-κB activation are critical events for septic AKI, it can be hypothesized that CK2α inhibition would be beneficial against septic AKI. To address this question, a cecum ligation and puncture (CLP)-induced AKI model was generated and investigated for the potential renoprotective effects of TBBt against septic AKI.

## 2. Results

### 2.1. CK2α Expression Is Increased in Kidney Tissues of Septic AKI Mice

Septic AKI was induced in C57BL/6J mice, and kidney and blood samples were obtained at various time points (Figure 1A). To determine whether CK2α is involved in septic AKI pathogenesis, protein levels were measured for CK2α in kidney tissues. Time-course analyses showed that the CK2α protein level reached peak levels at 3 h, continued to increase up to 6 h, and returned to the normal level at 24 h after CLP (Figure 1B).

### 2.2. CK2α Inhibition Increases Survival Outcome of Spetic AKI Mice

The survival of CLP mice with or without TBBt treatment was investigated next. Mortality was 80% within 48 h after CLP without TBBt treatment. However, in septic mice treated with 1 mg/kg of TBBt, the survival rate was 38%, and the survival rate was 55% when 2 mg/kg of TBBt was administered at 7 days after CLP (Figure 2).

### 2.3. CK2α Inhibition Protects Renal Function and Alleviates Kidney Injury

The renoprotective effects of TBBt were identified by measuring renal injury biomarkers, including serum creatinine and BUN. After 24 h of CLP, the levels of creatinine and BUN were elevated in CLP mice, while those levels were significantly reduced in TBBt-treated CLP mice (Figure 3A). The extent of renal injury was assessed via histological observation with H&E and PAS staining. In the CLP group, there was severe architectural disruption of the kidney, including tubular dilatation, brush border loss, and necrosis. However, less tubular injury and necrosis were observed in TBBt-treated mice (Figure 3B,C).

### 2.4. CK2α Inhibition Suppresses Apoptosis in Mice with Septic AKI

Although necrosis is the major cause of cell death in septic AKI, apoptotic cell death also contributes to histopathological changes in the kidney [13]. Therefore, we evaluated the extent of apoptosis via TUNEL staining. The number of TUNEL-positive apoptotic cells was considerably increased in CLP mice compared to sham mice (Figure 4A,B). Consistently, increased protein levels of the proapoptotic proteins cleaved caspase-3 and Bax, and decreased protein levels of the antiapoptotic protein Bcl-2 were observed in CLP mice (Figure 4C,D). Treatment with TBBt attenuated all of these changes.

### 2.5. CK2α Inhibition Decreases Inflammatory Responses in Mice with Septic AKI

Cytokines play an essential role in the initiation and progression of systemic inflammation in septic mice [4,5]. Therefore, we investigated the effect of TBBt as an inhibitor of inflammation during sepsis. The mice 12 h after CLP-induced sepsis had marked elevations in TNF-α, IL-6, and INF-γ levels in serum (Figure 5A). However, following sepsis induction, pretreatment with TBBt showed noticeably lower levels of these cytokines compared to CLP mice without TBBt. The cytokine levels were also analyzed in peritoneal fluid, and as with the results in the serum, the inhibitory effect of TBBt on cytokine levels was also confirmed in the peritoneal fluid.

Next, we determined F4/80, a specific surface marker of macrophages, in the kidney tissues of CLP mice. The results of the IHC staining showed that the number of F4/80^+^ macrophages was dramatically increased in CLP mice, whereas in TBBt-pretreated mice it was significantly decreased (Figure 5B).

### 2.6. CK2α Inhibition Suppresses MAPK-NF-κB Signaling Pathway in Mice with Septic AKI

To explore the mechanisms responsible for mediating the anti-inflammatory effect of TBBt, we measured NF-κB activation, which is a major signaling pathway of inflammation, in the kidney tissues of mice. Upon CLP, we observed the translocation of NF-κB subunits p50 and p65 into the nucleus (Figure 6A,B). In contrast, in the presence of TBBt, levels of p50 and p65 in the nucleus were reduced. Additionally, we observed an increase in the phosphorylation of IκB kinase (IKK) in CLP mice, whereas the ratio of p-IKK/IKK was decreased in TBBt-pretreated mice.

To investigate the mechanism by which TBBt suppresses the NF-κB signaling pathway, we compared the mitogen-activated protein kinase (MAPK) signal transduction pathways. We analyzed the lysates prepared from kidney tissues via Western blotting. Our results showed that CLP upregulated the phosphorylated forms of extracellular-signal-regulated kinase (ERK) and p38 MAPK, which were markedly suppressed by TBBt treatment (Figure 6C,D). These results suggest that TBBt suppresses inflammatory responses through the downregulation of the MAPKs–IKK–NF–κB axis in septic AKI mice.

## 3. Discussion

The aim of this study was to evaluate the potential effects of CK2α inhibition on septic AKI in mice. The results of this study clearly demonstrated that treatment with CK2α inhibitor TBBt alleviated sepsis-induced histopathological damage and improved renal function. Furthermore, TBBt not only increased the survival time but also the survival rate of septic mice. These beneficial effects were accompanied by decreases in inflammation and apoptosis. The underlying molecular mechanism appeared to involve the suppression of MAPKs and NF-κB activation.

Inflammation plays a critical role in the pathogenesis of septic AKI [14]. A recent study demonstrated that myeloid specific CK2α knockout mice are resistant to systemic bacterial infection [15], highlighting the importance of CK2α in inflammatory responses in an infectious bacterial setting. In agreement with this study, CK2α inhibition attenuated CLP-induced systemic inflammation, as evidenced by the decreased levels of cytokines (TNF-α, IFN-γ, and IL-6) in blood and peritoneum and also the reduced numbers of macrophages in kidney tissues. Macrophages are initiators of inflammation via the release of pro-inflammatory cytokines and mediators in response to endotoxin. Excessive macrophage infiltration directly affects renal parenchyma and promotes tubular cell apoptosis, which ultimately induces the occurrence of AKI [14]. The role of pro-inflammatory cytokines in sepsis-induced tissue damage has been well established. Pro-inflammatory cytokines such as TNF-α, IFN-γ, and IL-6 act as endogenous pyrogens, promoting the synthesis of other pro-inflammatory cytokines and inflammatory mediators by macrophages and mesenchymal cells. They are also known to stimulate the production of acute-phase proteins [16]. IL-6 levels are higher in patients who died from severe sepsis [17,18] and among pro-inflammatory cytokines, and plasma IL-6 levels have the best correlation with the mortality rate of sepsis patients [5], indicating that IL-6 is the key cytokine in the pathophysiology of severe sepsis. Plasma levels of TNF-α and IFN-γ are also markedly increased in patients with sepsis and in animal models [17,19]. Since the increase in macrophage infiltration and consequent release of pro-inflammatory cytokines appear to be an essential part of the pathogenesis in the inflammation process in septic AKI, this study provides a pharmacological basis for TBBt in the management of inflammation in septic AKI.

What is the underlying molecular mechanism by which TBBt exhibits an anti-inflammatory role? NF-κB is believed to be a master regulator of inflammation. Once activated, NF-κB initiates tissue injury by releasing a variety of pro-inflammatory cytokines and mediators. These released cytokines also activate NF-κB, which leads to a vicious cycle [17]. In this regard, several NF-κB inhibitors derived from either natural products or synthetic design have been proven to be highly effective for the treatment of septic AKI in mice [20,21,22,23]. In this study, increased phosphorylation of IKK in the cytosol and increased levels of NF-κB subunits in the nucleus were observed in the CLP mice, indicating that IKK-mediated phosphorylation of NF-κB subunits in the cytosol and concomitant nuclear translocation of NF-κB subunits. LPS-triggered MAPK activation is a critical part of signal transduction in modulating the activation of NF-κB [24]. This study demonstrates that TBBt treatment inhibits the phosphorylation of p38 MAPK and ERK and decreases NF-κB activity in the kidney tissues of CLP mice. In consistency with the suppression of pro-inflammatory cytokines (TNF-α, IFN-γ, and IL-6), it can be suggested that pretreatment with TBBt effectively relieves AKI, and its protective effects may be associated with the regulation of the MAPK-NF-κB-cytokines signaling pathway.

It is well known that the activation of NF-κB has been suggested to be positively correlated to apoptosis [25]. Additionally, the expression of caspase-3 is regulated by NF-κB in the kidney tissues of septic patients [26]. Consistent with these reports, pretreatment with TBBt decreased the protein levels of cleaved caspase-3 and the number of TUNEL-positive apoptotic cells in kidney tissues of CLP mice. This suggests that TBBt inhibits the activation of NF-κB, thereby exerting anti-apoptotic effects in septic AKI mice.

Apart from inflammation and apoptotic tissue injury, hemodynamic factors also play a critical role in the development and progression of AKI. Multiple mechanisms may cause microcirculatory dysfunction in septic AKI, such as endothelial injury, shedding of the glycocalyx, autonomic nervous system activation, and microthrombi formation [14]. In this regard, Ka et al. previously reported that TBBt treatment protected mice against bilateral renal ischemia–reperfusion injury. TBBt-treated mice presented preservation of renal function, histologically less tubular damage, and reduced infiltration of inflammatory cells [12]. However, an initial clinical trial with goal-directed therapy has shown little improvement in the mortality rates of septic AKI [27]. One explanation for these unexpected outcomes is that tissue damage might occur before microvascular alteration. Nevertheless, further study is required to figure out if TBBt could affect hemodynamic alterations in mice with septic AKI.

In the current study, a treatment regime of 2 mg/kg TBBt significantly attenuated the renal impairment. In a previous study, the same dose of TBBt could effectively protect against renal ischemia–reperfusion injury [12]. Because no noticeable adverse effects were not found in two studies, a 2 mg/kg dose of TBBt seems to be the most renoprotective dose in AKI animal models. This dose is less than other disease models. Intraperitoneal injection of 12.5 mg/kg of TBBt suppresses NF-κB activation in a murine model of asthma [28]. In a mouse model of bleomycin-induced dermal fibrosis, TBBt (2.5 mg/kg/day for three weeks) suppresses JAK2/STAT3 signaling [29].

In summary, a selective CK2 inhibitor, TBBt, alleviates sepsis-induced AKI by suppressing inflammation and apoptosis. Therefore, these results provide CK2 as a promising therapeutic target for treatment of sepsis-associated AKI.

## 4. Materials and Methods

### 4.1. Experimental Animals and Materials

Pathogen-free male 8-week-old C57BL/6J mice (body weight 20 ± 2 g) were purchased from Orient Bio (Sungnam, Republic of Korea). The mice were housed in a laminar airflow cabinet with a 12-h light/dark cycle and maintained on standard laboratory chow ad libitum. TBBt was purchased from Tocris (#2275, Bristol, UK) and dissolved in dimethyl sulfoxide (DMSO) to a concentration of 50 mg/mL. All other reagents were purchased from Sigma-Aldrich (St. Louis, MO, USA) unless otherwise noted.

### 4.2. Animal Model of Cecal Ligation & Puncture

To investigate the effect of TBBt during sepsis-induced AKI, mice were randomly divided into three groups: (i) sham-operated mice as the control group, (ii) the CLP group injected with normal saline, and (iii) the TBBt group with either 1 or 2 mg/kg in 200 μL of 5% DMSO in CLP-operated mice. TBBt was administered intraperitoneally once 3 h before CLP. CLP was performed as previously described elsewhere, with minor modifications [30]. Briefly, mice were anesthetized with a ketamine–xylazine mixture through an i.p. injection. To expose the cecum, the mouse abdomen was incised about 1 cm, and the cecum was ligated with a 4-0 silk suture (5 mm) from the base of the ileocecal valve. Next, the cecum was doubly perforated using a 24-gauge needle. A small amount of the bowel contents was extracted from both holes, and the cecum was inserted back into the abdomen. The skin of the abdomen was then closed, followed by a subcutaneous injection of resuscitative normal saline. Twenty-four hours after the CLP surgical procedure, mice were sacrificed via an overdose of sodium pentobarbital. Blood was collected from the hearts for biochemical analysis. Kidneys were collected, cut in half, and one half was immediately frozen in liquid nitrogen for protein analysis, while the other half was fixed in 4% formalin at room temperature for paraffin-embedded histological analysis. Another group of CLP mice with or without TBBt pre-treatment was observed to determine their survival rate for 7 days. All animal experiments were performed in accordance with the Guide for the Care and Use of Laboratory Animals, published by the US National Institutes of Health (NIH Publication No. 85-23, revised 2011). The current study protocol was approved by the Institutional Animal Care and Use Committee of Jeonbuk National University (Approval No. JBNU 2017-0088).

### 4.3. Western Blot Analysis

Kidney tissues were homogenized with proteinase and phosphatase inhibitors in a protein extraction solution (Intron Biotechnology, Burlington, MA, USA). The homogenates, which contained 20 μg of protein, were separated via 10% sodium dodecyl sulfate-polyacrylamide gel electrophoresis and transferred to polyvinylidene fluoride (PVDF) membranes. The blot was probed with 1:2500 diluted primary antibodies against CK2α (#2656), p-ERK (#4377), ERK (#4695), p-p38 MAPK (#4511), p38 MAPK (#8690), p-IKK (#2697), IKK (#2682), p50 (#3035), p65 (#8262), Bcl2 (#3498), cleaved caspase-3 (#9664), Bax (#5023, Cell Signaling, Beverly, MA, USA), lamin B1 (SC-374015, Santa Cruz Biotechnology, Dallas, TX, USA), and GAPDH (#A351, Bioworld Technology, St. Louis Park, MN, USA). Signals were detected with a Las-4000 imager (GE Healthcare Life Science, Pittsburgh, PA, USA). All the densitometry figures were obtained through Image J software (Version 1.53t).

Fractionation of the kidney tissue was performed using specific reagents for nuclear and cytosolic extraction (#78833, Thermo Scientific, Rockford, IL, USA). After centrifugation at 15,000× *g* for 10 min, the supernatants from the lysates were used for cytosolic extraction, while the pellets were used for nuclear extraction. All experiments were performed on ice.

### 4.4. Biochemical Analysis

Blood urea nitrogen (BUN) and serum creatinine were measured using specific assay kits (#K002-H1, Arbor Assays, Ann Arbor, MI, USA). TNF-α (#BMS607-3), IL-6 (#KHC-0061), and IFN-γ (#KMC4021, Invitrogen, Carlsbad, CA, USA) were measured using specific ELISA kits following the manufacturer′s instructions.

### 4.5. Histopathologic Assessment

The kidneys were collected and fixed with 4% formalin at room temperature for 24 h and were then paraffin-embedded. Tissues were cut into 4-μm thick sections, which were stained with hematoxylin and eosin (H&E) at room temperature and visualized under a microscope (Leica DM 2500; Leica Microsystems GmbH, Wetzlar, Germany). Histopathological damage was defined as swelling of the tubular epithelia, loss of brush border, degeneration of vacuole, necrosis of tubules, cast formation, and desquamation. The degree of tubular injury was estimated at a ×200 magnification using 5 randomly selected fields for each kidney according to the following standard: 0, normal; 1, damage involving <25% of tubules; 2, damage involving 25–50% of tubules; 3, damage involving 50–75% of tubules; and 4, damage involving 75–100% of tubules. For Periodic acid–Schiff (PAS) staining, sections were hydrated in ethanol and stained with a PAS reagent (#ab150680, Abcam, Cambridge, UK). Tubular necrosis was quantitated as the percentage of tubules in the outer medulla in which epithelial necrosis or necrotic debris was observed in PAS-stained sections.

### 4.6. Immunohistochemistry

Immunohistochemical staining was performed using the DAKO Envision system (Carpinteria, CA, USA), which utilizes dextran polymers conjugated with horseradish peroxidase to prevent contamination with endogenous biotin. After deparaffinization, tissue sections were subjected to a microwave antigen-retrieval procedure in a 10-mM sodium citrate buffer. Subsequently, endogenous peroxidases were blocked, and the sections were incubated with Protein Block Serum-Free (DAKO) to prevent non-specific staining. The sections were then immunostained with an antibody against F4/80 (#ab6640, Abcam) in 1:100 dilutions overnight. After washing with TRIS buffer saline (TBS, Sigma, St. Louis, MO, USA) 3 times for 10 min each, stained tissues were covered with a coverslip of an appropriate size. The stained sections were scanned under a microscope (Leica DM 2500) and quantified as the number of F4/80 positive cells per field using iSolution DT 36 software (Version 36.0) (Carl Zeiss, Oberkochen, Germany).

### 4.7. TUNEL Assay

TUNEL staining was used to detect apoptotic cells (#G3250, Promega, Madison, WI, USA). In brief, after treatment with a nucleotide mix and rTdT (terminal deoxynucleotide transferase), tissue sections were incubated at 37 °C for 1 h. Apoptotic cells, counterstained with hematoxylin, were covered with an appropriately sized coverslip. Apoptotic cells were counted under a microscope (×200) and expressed as the apoptosis index (number of apoptotic bodies/100 cells). Each group was assessed in triplicate, and the data were averaged.

### 4.8. Statistical Analysis

Data were expressed as the mean ± SD. Statistical comparisons were performed using one-way analysis of variance, followed by Bonferroni’s post hoc analysis. *p* values less than 0.05 were considered significant.

## Figures and Tables

**Figure 1 ijms-24-09783-f001:**
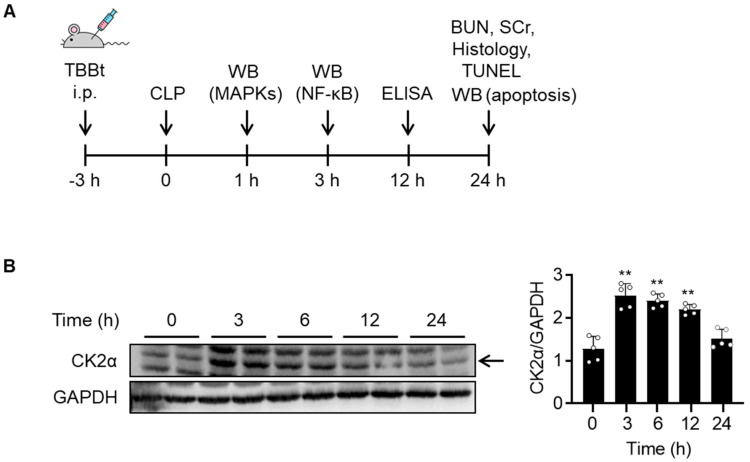
Alterations of CK2α expression in kidney tissues of septic AKI mice. (**A**) C57BL/6J mice were intraperitoneally injected with vehicle or TBBt and subjected to CLP. Kidney tissues and blood samples were collected at indicated times after CLP for each experiment. (**B**) Kidney tissues prepared from mice with CLP at the indicated time points were used to analyze CK2α expression. Protein intensity was measured. Values are mean ± SD (n = 5 mice per group). ** *p* < 0.01 vs. time 0. TBBt, 4,5,6,7-tetrabromobenzotriazole; CLP, cecum ligation and puncture; WB, Western blotting; ELISA, enzyme-linked immunosorbent assay; BUN, blood urea nitrogen; SCr, serum creatinine; TUNEL, terminal deoxynucleotidyl transferase dUTP nick-end labeling.

**Figure 2 ijms-24-09783-f002:**
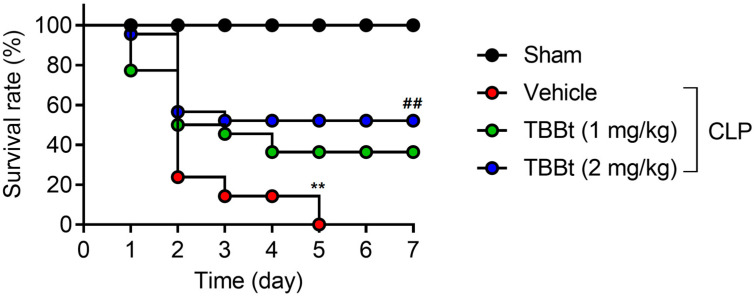
Improved survival outcome with CK2α inhibitor. Kaplan–Meier survival curves of CLP-operated mice treated with or without TBBt. Mice were randomly divided into three groups: sham, CLP, and CLP with TBBt (either 1 or 2 mg/kg). For the CLP plus TBBt group, TBBt was intraperitoneal administered 3 h before CLP. Other groups received the same volume of DMSO as a control. Survival was monitored for 7 d. Values are mean ± SD (n = 10 mice per group). ** *p*  <  0.01 vs. sham and ## *p*  <  0.01 vs. CLP + vehicle. TBBt, 4,5,6,7-tetrabromobenzotriazole; CLP, cecum ligation and puncture; DMSO, demethyl sulfoxide.

**Figure 3 ijms-24-09783-f003:**
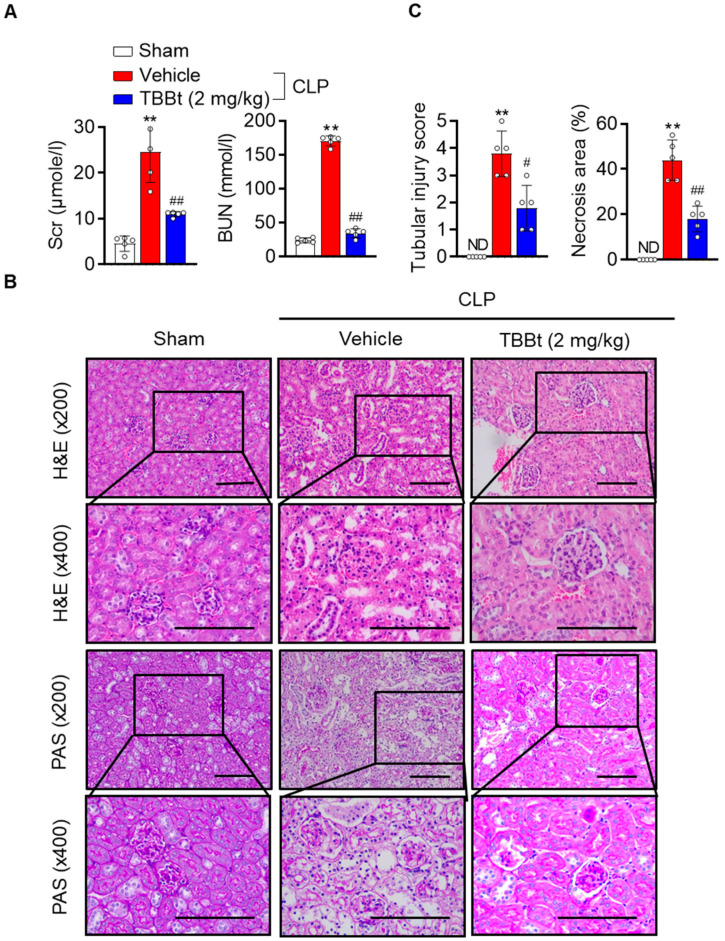
Reduced acute renal injury with TBBt. Mice were injected with 2 mg/kg of TBBt 3 h before CLP. (**A**) After 24 h CLP, blood samples were obtained for measurement of creatinine and BUN. (**B**) Kidney tissue were stained with H&E and PAS. Bars = 250 μm. (**C**) Histopathologic scoring and quantification of the necrotic area were performed in a blind fashion. The tubular injury score was given based upon epithelial simplification, tubular dilatation, vacuolization, and red blood cell (RBC) and hyaline casts (score 0: <1%, 1: ≥1~10%. 2: ≥10~25%, 3: ≥25~50%, 4: >50%). Bars show the mean ± SEM (n = 5). ** *p*  <  0.01 vs. sham; # *p*  <  0.05; and ## *p*  <  0.01 vs. CLP + vehicle. TBBt, 4,5,6,7-tetrabromobenzotriazole; CLP, cecum ligation and puncture; H&E, hematoxylin and eosin; PAS, periodic acid–Schiff.

**Figure 4 ijms-24-09783-f004:**
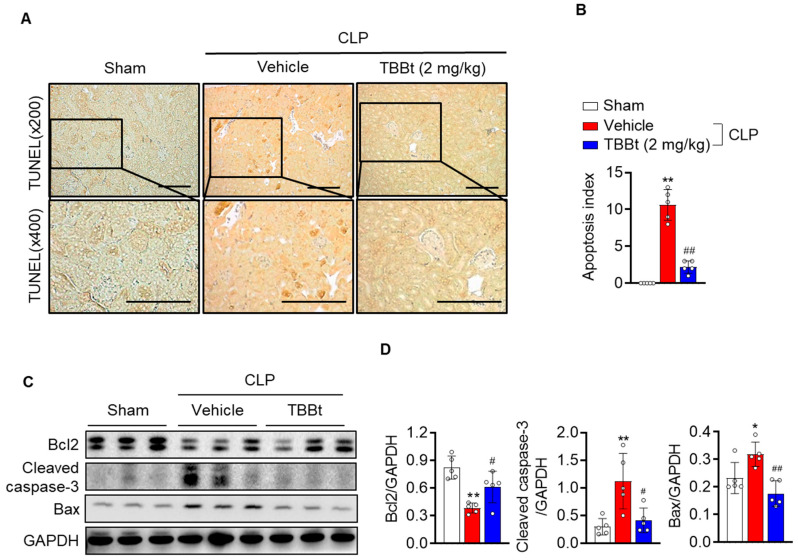
Suppression of CLP-induced apoptosis with TBBt. Mice were injected with 2 mg/kg of TBBt, and kidney tissues were collected 24 h after CLP. (**A**) Tissues were stained with TUNEL (×400). Bars = 250 μm. (**B**) Apoptotic cells were counted and expressed by a percentage which represented all glomerular and tubular cells. (**C**,**D**) The expression levels of Bcl-2, cleaved caspase-3, and Bax were examined via Western blotting at 24 h after CLP. Protein intensity was measured. Values are expressed as the mean ± SD (n = 5 mice per group). * *p* <  0.05 and ** *p* <  0.01 vs. sham; # *p* <  0.05; and ## *p* <  0.01 vs. CLP + vehicle. TBBt, 4,5,6,7-tetrabromobenzotriazole; CLP, cecum ligation and puncture; TUNEL, terminal deoxynucleotidyl transferase dUTP nick-end labeling.

**Figure 5 ijms-24-09783-f005:**
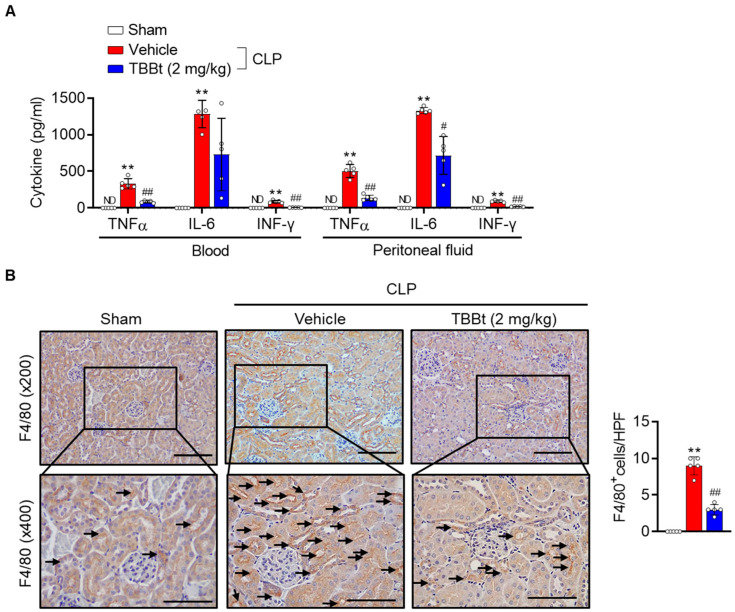
Decreased inflammatory responses with TBBt. Mice were injected with 2 mg/kg of TBBt 3 h before CLP. (**A**) Protein levels of inflammatory cytokines in blood and peritoneal fluid were measured via ELISA 12 h after CLP. (**B**) Immunohistochemistry to identify infiltrating macrophages (F4/80^+^) at 24 h after CLP. Bars = 50 μm. F4/80^+^-positive macrophages (indicated with arrows) were counted in at least 5 photographs at ×400 magnification per animal and expressed as the number of positive macrophages per high-power field (HPF). Values are mean  ±  SD (n = 5 mice per group). ** *p* < 0.01 vs. sham; # *p* < 0.05; and ## *p* < 0.01 vs. CLP + vehicle. TBBt, 4,5,6,7-tetrabromobenzotriazole; CLP, cecum ligation and puncture; ND, not detected.

**Figure 6 ijms-24-09783-f006:**
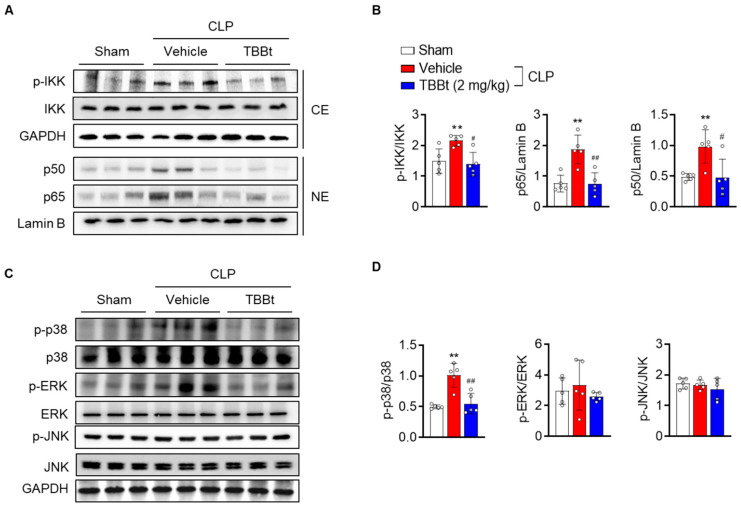
Suppression of MAPKs and NF-κB pathways in kidney tissues of septic mice with TBBt. (**A**,**B**) Mice were injected with 2 mg/kg of TBBt 3 h before CLP. Cytosolic (CE) and nuclear extract (NE) were prepared from kidney tissues 3 h after CLP. Nuclear translocation of p50 and p65 subunits and phosphorylation of cytoplasmic IKK were analyzed via Western blotting (n = 5 mice per group). (**C**,**D**) Total lysates prepared from kidney tissues 1 h after CLP were used to determine the total and phospho-forms of p38 MAPK and ERK. Protein intensity was measured. Values are mean ± SD (n = 5 mice per group). ** *p* < 0.01 vs. sham; # *p* < 0.05; and ## *p* < 0.01 vs. CLP + vehicle. TBBt, 4,5,6,7-tetrabromobenzotriazole; CLP, cecum ligation and puncture.

## Data Availability

Not applicable.

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
