# Peer review of "Casein Kinase 2 Alpha Inhibition Protects against Sepsis-Induced Acute Kidney Injury"

_ijms, 2023, doi:10.3390/ijms24129783_

Round 1
Reviewer 1 Report
In the current study titled “Casein Kinase 2 alpha inhibition protects against sepsis-induced acute kidney injury” by Jeung-Hyun Koo et.al. authors have addressed Reno-protective role of CK2 inhibition in sepsis induced AKI. The have utilized CLP model and further showed mechanistic correlation with signaling proteins correlation. It is an interesting article to support TBBt, a CK2 inhibitor can protect renal damage post AKI.
The manuscript is well organized in terms of relevant literature, background, results, and discussion. However, I have certain questions/suggestions to make the results more evident:
1- Firstly, SCr expressed value doesn’t corelate renal damage, probably some calculation mistake and needs revision as it should be 10 folds higher at least to show any renal injury.
2- Also, for SCr, I am interested to see data with individual mice values displayed on graph instead of bar graph.
3- Figure 5 B, n=3 is too small to draw a statistical difference. At least, n=5 would make it stronger representation. Highlighting the macrophage population with arrow head might make the picture more convincing.
4- Again, for your histopathology/fibrosis data the tubular injury in first panel figure 3B doesn’t justify the effect of TBBt as represented in graph. Please, get a better picture for lower magnification.
In general, more clarity is needed on scientific methods and findings to support the results.
I am interested to know if there are any future studies planned for further evaluation as, n=10 was chosen for initial part of the study but most of the data down the manuscript is representing n=3.
Discussion and References:
All relevant and in sync.
Author Response
In the current study titled “Casein Kinase 2 alpha inhibition protects against sepsis-induced acute kidney injury” by Jeung-Hyun Koo et.al. authors have addressed Reno-protective role of CK2 inhibition in sepsis induced AKI. They have utilized CLP model and further showed mechanistic correlation with signaling proteins correlation. It is an interesting article to support TBBt, a CK2 inhibitor can protect renal damage post AKI.
The manuscript is well organized in terms of relevant literature, background, results, and discussion. However, I have certain questions/suggestions to make the results more evident:
1- Firstly, SCr expressed value doesn’t corelate renal damage, probably some calculation mistake and needs revision as it should be 10 folds higher at least to show any renal injury.
Response: Based on the definition and classification of AKI by KDIGO, AKI can be described as an increase in serum creatinine (SCr) of more than 0.3 mg/dL (26.5 μmol/L) within 48 hours, or an increase in SCr greater than 1.5 times the baseline within a week. Our findings following CLP (cecal ligation and puncture) align with the classification of AKI. Additionally, our SCr results indicate AKI stage III, as they surpass three times the baseline and exceed SCr levels of 4.0 mg/dL (354 μmol/L). Despite the seemingly modest nature of our SCr results, they are significant enough to warrant classification as severe AKI stage. Our study's outcomes are also consistent with those of other AKI studies.
Reference: Khwaja, A: KDIGO Clinical Practice Guidelines for Acute Kidney Injury. Nephron Clinical Practice 2012, 120(4): c179-c184.
2- Also, for SCr, I am interested to see data with individual mice values displayed on graph instead of bar graph.
Response: As commented, we have included the individual values on the graph.
3- Figure 5 B, n=3 is too small to draw a statistical difference. At least, n=5 would make it stronger representation. Highlighting the macrophage population with arrow head might make the picture more convincing.
Response: As per the reviewer's comment, we conducted additional analysis on more mice (n=5) in Figure 5B. The authors have also addressed the reviewer's comment by changing the pictures of the macrophage population, indicating them with arrowheads.
4- Again, for your histopathology/fibrosis data the tubular injury in first panel figure 3B doesn’t justify the effect of TBBt as represented in graph. Please, get a better picture for lower magnification.
Response: We have included a picture at a lower magnification, as per the reviewer's comment.
In general, more clarity is needed on scientific methods and findings to support the results.
I am interested to know if there are any future studies planned for further evaluation as, n=10 was chosen for initial part of the study but most of the data down the manuscript is representing n=3.
Response: We have enhanced the description of the Materials and Methods to provide more detail. Furthermore, we re-analyzed a larger number of kidney slides (n=5).
Reviewer 2 Report
I think this is an interesting and promising study with good results. The following points represent flaws and need to be adressed:
1. Abstract should be written again in order to get more scientific soundness.
2. Material and methods shold be written in more detail in order to achieve reproducibility. At first, number of animals should be added (number of all used animals, number of animals which samples were used for analysis, number of animals that were used for monitoring and determination of their survival rates). Also, the authors should add in material and methods intraperitoneal sampling procedure, biochemical analysis in more datail, concentration of antibodies used for immunochistochemical procedure.
3. In the results section, all histophatological and immunochistochemical results should be presented with bigger and more visible pictures, marked and explained in more detail. The reader cannot see any result nor draw conclusions from presented pictures (Figure 3,4,5). Authors should sepatate the pictures, show bigger pictures and mark the changes between the grups with arroews or signs and to explaine all of that in figure legends.
4. At the end, I believe that the biggest problem and limitation of this study is the small sample size and significance obtained on such a small sample (n=3 for mostly all performed analysis). It is too small sample for valid ANOVA comparisons, especially considering large SD obtained for some parameters in some groups. Also, Fisher is not the most accurate post hoc test, because unlike the Bonferroni, Tukey, Dunnet and Holm post hoc tests does not correct for multiple comparisons. Authors should add more animal samples per group to obtaine valid statistics and valid interpretation of the results.
Minor editing of English language required.
Author Response
I think this is an interesting and promising study with good results. The following points represent flaws and need to be addressed:
1. Abstract should be written again in order to get more scientific soundness.
Response: The authors have made revisions to the Abstract section in response to the reviewer's comment.
2. Material and methods should be written in more detail in order to achieve reproducibility. At first, number of animals should be added (number of all used animals, number of animals which samples were used for analysis, number of animals that were used for monitoring and determination of their survival rates). Also, the authors should add in material and methods intraperitoneal sampling procedure, biochemical analysis in more datail, concentration of antibodies used for immunochistochemical procedure.
Response: We have provided a more detailed description of the Materials and Methods.
3. In the results section, all histophatological and immunochistochemical results should be presented with bigger and more visible pictures, marked and explained in more detail. The reader cannot see any result nor draw conclusions from presented pictures (Figure 3,4,5). Authors should separate the pictures, show bigger pictures and mark the changes between the grups with arroews or signs and to explaine all of that in figure legends.
Response: We have provided larger images in the figures, highlighting the changes in greater detail, and included explanations in the figure legends. In Figure 3, we have replaced the images in Figure 3B. We have added a low magnification (×200) image of PAS to depict structural changes, damaged glomeruli, and tubular membrane (Figure 3B). Additionally, we have re-analyzed a higher number of kidney slides (n=5) and re-evaluated the tubular injury score and necrotic area (Figure 3C). We also described how we obtained the tubular injury score in the figure legend.
In Figure 4, we have replaced the images in Figure 4A, 4B and 4C. We added a high magnification (×400) image of TUNEL to accurately quantify the apoptotic injured area by analyzing the DNA fragments (Figure 4B). As a result, we re-analyzed a higher number of kidney slides (n=5) and re-evaluated the apoptosis index (Figure 4B). We also replaced western blot results of Bax (Figure 4C) with a better picture.
In Figure 5, we have included a lower magnification (x200) image of F4/80 (Figure 5B) and marked arrows to indicate the presence of macrophages. Additionally, we re-analyzed a higher number of kidney slides (n=5). We assessed macrophage infiltration and confirmed a reduction in the number of macrophages, as well as a less damaged membrane structure in the TBBt group. Furthermore, we observed that CK2 inhibitor ameliorated macrophage-induced kidney damage.
4. At the end, I believe that the biggest problem and limitation of this study is the small sample size and significance obtained on such a small sample (n=3 for mostly all performed analysis). It is too small sample for valid ANOVA comparisons, especially considering large SD obtained for some parameters in some groups. Also, Fisher is not the most accurate post hoc test, because unlike the Bonferroni, Tukey, Dunnet and Holm post hoc tests does not correct for multiple comparisons. Authors should add more animal samples per group to obtaine valid statistics and valid interpretation of the results.
Response: In response to the reviewer's comment, we conducted additional analysis on a greater number of mice (n=5) in Figure 5B. As per the reviewer's comment, the authors have also modified the pictures of the macrophage population, indicating them with arrows. Additionally, we performed a one-way analysis of variance and applied the Bonferroni post hoc test, as recommended by the reviewer.
Round 2
Reviewer 1 Report
I appreciate authors for addressing and making suggested changes. In total it enhanced the quality of manuscript overall presentation.
Reviewer 2 Report
The authors successfully responded to all raised objections.
Minor editing required.